# Development and Psychometric Properties of the Health Belief Scales toward COVID-19 Vaccine: A Cross-Sectional Study in North-Eastern Poland

**DOI:** 10.3390/ijerph19095424

**Published:** 2022-04-29

**Authors:** Karol Konaszewski, Jolanta Muszyńska, Sebastian Binyamin Skalski, Janusz Surzykiewicz

**Affiliations:** 1Faculty of Education, University of Bialystok, 15328 Bialystok, Poland; jolamusz@uwb.edu.pl; 2Institute of Psychology, Polish Academy of Sciences, 00378 Warsaw, Poland; sebastian.skalski@sd.psych.pan.pl; 3Faculty of Philosophy and Education, Katholische Universität Eichstätt-Ingolstadt, 85072 Eichstätt, Germany; janusz.surzykiewicz@ku.de; 4Faculty of Education, Cardinal Stefan Wyszynski University in Warsaw, 01938 Warsaw, Poland

**Keywords:** COVID-19 vaccine, health belief model, Poland

## Abstract

In the study, we evaluated the psychometric properties of the Health Belief Scales Toward COVID-19 Vaccine, including the structure, reliability and validity of the scale. Psychometric properties were assessed on a general sample of 472 Polish participants aged between 19 and 69 years (M = 25.43). The procedure consisted of completing the Health Belief Scales Toward COVID-19 Vaccine, the World Health Organization’s 5-item Well-being Index (WHO-5) and demographic questions. The presented research results obtained using the measure indicate that it can be considered to be a reliable and valid research tool. A four-factor solution demonstrated a good fit to the data: χ2/df = 3.90, RMSEA = 0.079, AGFI = 0.913, GFI = 0.951, CFI = 0.960, TLI = 0.941. The reliability measures for the overall index of the Health Belief Scales Toward COVID-19 Vaccine were: Cronbach’s α = 0.88 and McDonald’s ω = 0.87. The Health Belief Scales Toward COVID-19 Vaccine total score correlated negatively and weakly with the WHO-5 score. The Health Belief Scales Toward COVID-19 Vaccine proved to be a valid and a reliable tool to assess attitudes towards vaccination in four dimensions consistent with the HBM.

## 1. Introduction

The coronavirus that causes COVID-19 disease was first detected in Wuhan City on 31 December 2019. It then quickly spread across the world, and on 11 March 2020 the World Health Organization (WHO) declared a state of pandemic. By January 2022, the total number of positive cases had exceeded 275 million. The number of reported deaths had reached a figure of more than 5 million. Thus, the COVID-19 pandemic was considered to be the most serious global threat to humanity since World War II. Because of the lack of effective treatments, the mass use of COVID-19 vaccines has been the key strategy for containing the pandemic [1,2,3].

Vaccination is recognized as the most effective and cost-effective public health intervention, and has made a very large contribution to improving global health by reducing morbidity and mortality from many infectious diseases. Poland and the whole world are currently experiencing successive waves of the COVID-19 pandemic; thus, it seems particularly important to achieve herd immunity by properly vaccinating the population against COVID-19. According to recent studies, in order to achieve herd immunity, it is necessary to vaccinate between 55% and 82% of the population [4,5,6,7]. Although vaccines are available free of charge throughout the European Union (as they are in the United States), the percentage of fully vaccinated people is, in some countries or regions, less than 50%, and this is the case in Poland. Poland ranks 23rd in the European Union (as of 20 January 2022) in terms of the number of people who have received at least one dose of the COVID-19 vaccine, and therefore has a low figure compared to other Member States (Data shown are for the period in which the study was conducted, source: https://www.who.int/covid-19/vaccines, accessed on 20 January 2022). In this situation, in addition to vaccine supply, it seems important to understand the psychological mechanisms underlying the formation of attitudes towards COVID-19 vaccination. Exploring beliefs about vaccination and possible factors influencing willingness to be vaccinated when vaccination rates are low may increase the motivation and number of decisions made to receive a COVID-19 vaccination [8,9]. 

In research on vaccination attitudes, particular attention is given to the Health Belief Model (HBM), which was developed to understand people’s failure to adopt disease prevention strategies or to undergo screening tests for the early detection of disease [10,11,12,13]. Later applications of the HBM looked at patients’ responses to symptoms and adherence to treatment. The HBM suggests that a person’s belief in their personal risk of disease, along with their belief in the effectiveness of recommended health behaviours or actions, predict the likelihood that they will adopt the behaviour. The HBM has been used to assess beliefs and attitudes towards seasonal and pandemic swine flu vaccines [14,15], and to assess the association between beliefs and hepatitis B vaccinations purchased by the individual [16,17]. The HBM has been used extensively in studies that have predicted the adoption of preventive and health behaviours [18], engagement in physical activity to improve health [19], and perceived barriers and benefits of weight management [20]. In addition, current reports indicate positive associations between health behaviours and mental wellbeing [21]. Research has also demonstrated that trust in the health care system and vaccine manufacturer is a key component of health education programmes to promote life-saving vaccines [22]. 

The HBM includes four factors: perceived susceptibility and perceived severity (beliefs about disease and beliefs about the consequences of getting sick), perceived benefits (beliefs about the benefits the individual will receive from participating in health behaviours), barriers to vaccination (any barriers that may stand in the way of vaccination), and cues to action (exposure to information that encourages action). The model has been successfully used to explain why individuals engage in vaccination-related health behaviours, and can be used to design programmes to address problems related to improving vaccination beliefs in COVID-19 [11,15,23]. In this framework, there is, to date, no specific and widely available drug for COVID-19, while the virus continues to spread widely worldwide, and patients more than 65 years old and those living with comorbidities remain particularly at risk of severe illness and death [1,2,3,4,5]. Thus, immunization appears to be an important health-promoting behaviour to improve public health. Today, the development of vaccines against COVID-19 gives us the only hope of reducing morbidity and mortality due to COVID-19 and thus halting the development of the pandemic. Nevertheless, vaccine beliefs have been one of the main barriers to vaccine uptake and the achievement of herd immunity, especially for the protection of the most vulnerable populations. In view of this, the HBM has been widely used to gain accurate insights into people’s health behaviours, and plays an important role as a predictor of vaccine uptake, as has been demonstrated in previous studies [13,23].

People of all ages can become infected with the virus that causes COVID-19 disease, and the elderly and people with chronic illnesses are at high risk for severe illness and death. Acceptance and adoption of new vaccines is a significant and unprecedented challenge. In previous studies, the HBM has proven to be a useful tool for predicting vaccination behaviour [10,11,12,13,14,15,16,17,18,19,20,21,22]. Given the assumptions of the model, Huynh and colleagues [24] developed the Health Belief Scales Toward COVID-19 Vaccine. The scale has proved to be a valid and reliable diagnostic tool that can be helpful for health educators to assess vaccination beliefs among both individuals, communities, and society at large when there is a need to implement new vaccines in cases, such as that of the COVID-19 vaccine, before the vaccines are more widely used throughout the community. The scale consists of 12 items that address four factors: disease threat, perceived benefits, perceived barriers, and cues to action. They explain 68.3% of the total variance of the explained variable. Confirmatory factor analysis using index modification has demonstrated a good fit of the four-factor model to the theoretical assumptions of the HBM. To our knowledge, there has been no valuable and reliable tool using the HBM to assess individuals’ beliefs about COVID-19 vaccines in the Polish population.

For decades, vaccines have been an effective means of preventing disease. However, vaccination hesitancy and vaccine refusal are a serious problem globally, prompting the World Health Organization (WHO) to rank this uncertainty as one of the top 10 health threats in 2019. According to various studies, reasons for vaccination hesitancy include religious considerations, personal beliefs, and safety concerns due to widely held myths [25,26]. Unfortunately, to the best of our knowledge, no studies assessing attitudes toward vaccination have been conducted in Poland to date.

The objective of this study is to evaluate the psychometric properties of the Health Belief Scales Toward COVID-19 Vaccine on a sample of unvaccinated Polish people, including the structure, reliability and validity of the scale. The study tested the four-factor solution proposed by Huynh and colleagues [24], which is consistent with the HBM theoretical model. The validity of the scale was assessed by examining the relationship between belief in COVID-19 vaccination and well-being. We hypothesised that the association between the variables would be negative. The study sought to answer the following research question: What are the psychometric properties of the Polish version of the Health Belief Scales Toward COVID-19 Vaccine? The data obtained on attitudes related to COVID-19 vaccination in the unvaccinated population may contribute to the development of methods of influence in changing these attitudes and securing health.

## 2. Materials and Methods

### 2.1. Participants and Procedure

Psychometric properties were assessed on a general sample of 472 Polish participants aged between 19 and 69 years (M = 25.43, SD = 7.30), 69% of whom were women. Participants were differentiated according to the declaration of financial situation (12% very good, 40% good, 29% average, 16% bad, and 3% very bad) and place of residence (30% rural, 70% urban). Among the survey participants, 36% were gainfully employed, 34% were gainfully employed and studying, 25% were only studying, and 6% were unemployed. All data obtained from respondents were complete. Demographic characteristics of the sample are detailed in Table 1.

The study was conducted in north-eastern Poland. The University of Bialystok database was used to send information about the study. The database contained e-mails to 12,000 students and graduates who agreed to have their e-mail addresses processed for research purposes. To avoid selection bias, the invitation to participate in the study was sent to all individuals in the database. In order to obtain reliable information on barriers to vaccination, the criterion for entry into the study was the declaration that one is an unvaccinated person. The survey was conducted on non-vaccinators because it seems crucial to know their views in developing educational implications aimed at health prevention. The study was conducted in November and December 2021 with the approval of the Ethics Committee of the Institute of Psychology, Polish Academy of Science. The participants were informed that: (1) participation in the study was voluntary; (2) they could decline to participate at any time during the study; (3) the study was anonymous; (4) individual responses would not be shared with anyone; and (5) aggregate results would be analysed and used for research purposes only. The Google Forms platform was used to collect data. Respondents gave informed consent and were informed of the objectives and procedure. All procedures performed in the study involving the human participants complied with the ethical standards of the Ethics Committee of the Polish Academy of Science. The procedure consisted of completing the Health Belief Scales Toward COVID-19 Vaccine, the World Health Organization’s 5-item Well-being Index (WHO-5) and demographic questions.

In the process of the Polish adaptation of the scale, in accordance with WHO [27] standards for adaptation and validation procedures for psychological research tools, the scale was translated into Polish in line with the WHO recommendations through (1) forward-translation, (2) expert panel back-translation, (3) pre-testing and cognitive interviewing, and (4) production of the final version. Two independent translators were engaged to translate the original English language version. In the next step, an expert panel was appointed consisting of four people fluent in English and specializing in health psychology and psychological research methodology. Taking into account the design of the questionnaire and the research model of the original tool, the version of questions best fitting the theory and the specificity of Polish conditions was selected. In the next step, two other independent translators were commissioned to re-translate the questionnaire items in order to check the accuracy of the translation and its fidelity to the original text. As a result of this analysis of the translation, problematic statements were corrected to produce an unambiguous version of the translation. The final Polish version of the scale was approved by the expert panel.

### 2.2. Measures

**The Health Belief Scales Toward COVID-19 Vaccine** is the subject of this adaptation. The scale is designed to measure beliefs about vaccination. The scale consists of 12 self-descriptive statements. The participants express their attitude towards each of the statements on a 5-point Likert scale, from 1 = “I strongly disagree” to 5 = “I strongly agree”. The four domains of the scale are: (1) perceived threat, which combines susceptibility and severity, (2) perceived benefits, which is what the respondent knows about the benefits that they will get from receiving the vaccination (knowledge of the benefits of vaccination), (3) barriers to vaccination, or what limits and hinders vaccination, and (4) cues to action, which are external events that prompt a desire to change behaviour. In addition, the study can determine an overall scale index by summing the scores obtained across all statements. The internal consistency, measured with a standardized Cronbach’s alpha coefficient, was 0.76 [24]. The scale is based on the HBM model, which has been demonstrated to be effective in predicting COVID-19 vaccination attitudes in the Vietnamese population. Overall, it is demonstrated that the vaccination COVID-19 scales based on HBM had a sufficient factor validity by most of the satisfactory psychometric results of the tests. The scale can be a helpful tool for health educators to assess vaccination beliefs among individuals and the community in situations where there is a need to implement new vaccines before they are more widely used throughout the population. This is a novel and valid tool to assess an individual’s beliefs about the COVID-19 immunization (see [24]).

**The World Health Organization’s 5-item Well-being Index (WHO-5)** was used to measure well-being. The scale consists of five self-descriptive statements. The participants express their attitude towards each of the statements on a 6-point Likert scale, from 0 = “At no time” to 5 = “All of the time” (in relation to the past two weeks). The Polish version of the WHO-5 is characterized by satisfactory psychometric properties (α = 0.87). The scale includes statements: ‘I have felt cheerful and in good spirits’; ‘I have felt calm and relaxed’; ‘I have felt active and vigorous’; ‘I woke up feeling fresh and rested’; and ‘My daily life has been filled with things that interest me’ [28].

### 2.3. Data Analysis

A “sampling calculator” was used to calculate the sample size. A 95% confidence level, a fraction size of 0.5, and a maximum error of 5% were used. The total number of respondents is 12,000. Analysis with the abovementioned parameters would demand a sample of at least 372 participants.

Confirmatory factor analysis (CFA) was used to determine the factor structure. Structural equation analysis was performed using the AMOS program. The model parameters were estimated using the maximum likelihood method (ML). The following indicators were used to assess the goodness of fit of the model to the data: GFI (goodness of fit index), AGFI (adjusted goodness of fit index), RMSEA (root-mean-square error of approximation) TLI (Tucker–Lewis index), CFI (comparative fit index), and chi-squared test (χ2/df). AGFI and GFI values of at least 0.90 indicate a good and adequate fit of the model to the data [29]. The criterion for acceptance of the χ2/df value varies across researchers. An acceptable value of χ2/df is less than 5 [30]. An RMSEA value of less than 0.08 can also be interpreted as showing a good fit to the data [31,32]. Hu and Bentler [29] suggest that values for CFI and TLI greater than 0.95 generally indicate a relatively good fit between the model and the data. As regards to the refinement of the model, the model fit indices may approach the thresholds mentioned above but not come close enough to be considered satisfactory. If this is the case, minor adjustments can be made to the relationships in the model, and then the model can be tested again. The determination of which adjustments to make can be guided by the use of modification indices, which provide an estimate of the improvement in model fit that will occur by adding a given relationship, including direct paths and correlations [33]. When modifications are added to the model, the model is restarted and interpreted with the new fit indices.

The reliability of the scale was calculated using Cronbach’s α and McDonald’s ω coefficients. A generally accepted rule of thumb is that an α of 0.6–0.7 indicates an acceptable level of reliability [34,35]. Student’s *t*-test for independent samples was used to assess potential differences for sex and place of residence. One-way analysis of variance (ANOVA) was used to compare means across groups (financial situation and work situation). Pearson’s correlation analysis was used to determine the relationships between the variables. Following Cohen [36], the absolute value of a correlation is equivalent to its effect size, with values under 0.10 representing a trivial effect, those between 0.10 and 0.30 a small/weak effect, those between 0.30 and 0.50 a medium effect and those above 0.50 a large effect. Effect sizes were assessed using Cohen’s d. Effects with the d of 0.20 to 0.50 were interpreted as small, effects with d of 0.50 to 0.80 were considered medium, and effects with d greater than 0.80 were considered large [36]. The Item Difficulty Index (IDI) was used to assess the ceiling and floor effects. A ceiling effect was deemed to be observed when the IDI was greater than 0.8, and a floor effect when the IDI was less than 0.2. Data granulation was concluded when over 50% of the respondents gave the same answer.

## 3. Results

### 3.1. Item Analysis

Table 2 presents the basic descriptive statistics for the statements analysed. The IDI values did not show the existence of floor effects or ceiling effects in the data. Analysis of the frequency of responses for each test item revealed no problems with data granularity. Therefore, all items were included for further analysis.

### 3.2. Assessment of Structure and Reliability 

We used CFA to evaluate the scale structure. In the first step, we confirmed the model proposed by the scale authors [24]. Model fit was found for the four HBM dimensions. The factor loadings of 11 of the items were good, and ranged from 0.50 to 0.92. Item 10, “Uważam, że w przyszłości aby się zaszczepić koszt szczepionki na COVID-19 będzie wysoki [I think that the COVID-19 vaccine will be expensive]” had a low discriminatory power of less than 0.30. The model analysed was found to be an acceptable fit to the data: χ2/df = 4.02, RMSEA = 0.080, AGFI = 0.899, GFI = 0.930, CFI = 0.947, TLI = 0.926. To fit the model to the data better, we decided to remove item 10 from the model. We also tracked potential modification indices that could improve the fit of the four-factor model to the data. The modified model with covariates between items such as “5. Szczepienie uchroni mnie przed zachorowaniem na COVID-19 [Immunization will prevent me from contracting COVID-19]” and “7. Dzięki zaszczepieniu będę mieć mniejsze obawy o ciężki przebieg COVID-19 [By being immunized, I feel less worried about the possibility of severe illness from getting COVID-19]” resulted in improved fit of the statistics to the data: χ2/df = 3.90, RMSEA = 0.079, AGFI = 0.913, GFI = 0.951, CFI = 0.960, and TLI = 0.941. Thus, the revised model produced a better fit to the data than the original model by increasing the GFI, AGFI, CFI, and TLI values and decreasing the RMSEA and χ2/df values. Figure 1 shows the standardized solution for the revised four-factor model.

### 3.3. Reliability of the Index Created

The reliability measures for the overall index of the Health Belief Scales Toward COVID-19 Vaccine were: Cronbach’s α = 0.88 and McDonald’s ω = 0.87. The reliability measures of the factor of perceived susceptibility and perceived severity were α = 0.73, ω = 0.72. The reliability measures of the factor of perceived benefits were α = 0.88, ω = 0.89. The reliability measures of the factor of perceived barriers were α = 0.71, ω = 0.71, and the reliability measures of the factor of cues to action were α = 0.81, ω = 0.81. Because each subscale consists of between two and four statements for each factor, this reliability should be considered good [37,38]. 

### 3.4. Descriptive Statistics of the Health Belief Scales toward COVID-19 Vaccine and Relations Based on Demographic Variables

To complete the data analysis, calculations were performed on the descriptive statistics for the COVID-19 vaccine belief total score and the four subscales. These results are summarised in Table 3. Using the Student’s t-test analysis for independent samples, we examined whether there was a relationship between the COVID-19 vaccination belief score we had created and the sex of the respondent. We confirmed that there were differences between women on the perceived susceptibility and perceived severity index (*t*_(470)_ = −2.45, *p* < 0.05; Cohen’s *d* = 0.24). For the other factors, no statistically significant differences were found between men and women. People from rural areas had higher level of perceived barriers index than people from cities (*t*(470) = −2.04, *p* < 0.05; Cohen’s *d* = 0.20). Place of residence did not differentiate the other indicators in a statistically significant manner.

One-way analysis of variance (ANOVA) did not confirm that the financial situation and work situation differentiated the results for beliefs about receiving the COVID-19 vaccine and the four component factors of these beliefs in a statistically significant manner. 

The overall vaccination belief score was associated with age (*r* = 0.13; *p* < 0.01). This correlation was positive and weak. Moreover, the correlations between age and perceived susceptibility and perceived severity (*r* = 0.15; *p* < 0.01) and age and cues to action (*r* = 0.09; *p* < 0.05) were weak and positive. However, there were no significant associations between age and perceived benefits or age and perceived barriers.

### 3.5. Convergent Validity 

Validity was assessed by evaluating the correlation coefficients with the WHO-5 score. We expected a negative correlation between the constructs. The Health Belief Scales Toward COVID-19 Vaccine total score correlated negatively and weakly with the WHO-5 score (r = −0.17, *p* < 0.01). The WHO-5 score also correlated negatively and weakly with the four factors, namely perceived susceptibility and perceived severity (r = −0.16, *p* < 0.01), perceived benefits (r = −0.10, *p* < 0.05), perceived barriers (r = −0.12, *p* < 0.01), and cues to action (r = −0.15, *p* < 0.01).

### 3.6. Method of Score Calculation

Finally, the Health Belief Scales Toward COVID-19 Vaccine include the following items: 1, 2, 3, 4 (perceived susceptibility and perceived severity); 5, 6, 7 (perceived benefits); 8, 9 (perceived barriers); and 11, 12 (cues to action). With the accepted method of response, statements 8 and 9 are reversed. Since the subscales have different numbers, we suggest that the mean rather than the sum should be calculated for each subscale.

## 4. Discussion

Vaccination is associated with important resources and social values, such as health and safety, social responsibility, and trust in science [9,11,14,26]. Therefore, the understanding of vaccination beliefs can be a real scientific challenge, and the development of a tool for this is a necessity and a foundation for educational and social action [23,25]. To the best of our knowledge, no reliable tool to assess COVID-19 vaccine beliefs based on the HBM has existed in Poland to date. All the items included in the adapted scale were based on the four factors of the HBM theoretical model and were accepted by experts according to the expected content validity. The CFA demonstrated that the four-factor model proposed by Huynh and colleagues [24], which included error modification between items 5 and 7 and the removal of item 10, was a very good fit to the data. Item 10 had low discriminatory power because it related to the high cost of the vaccine. It should be noted that in Poland vaccines are completely free, and there is no indication that there will be a charge for them in the future, which seems to explain the low association between the statement and the overall subscale score. After removing item 10, both the total score and the four dimensions comprising the scale had good internal reliability [37], as assessed by the Cronbach’s alpha and McDonald’s omega coefficients.

We assessed the validity of the tool by analysing the relationship between vaccination beliefs and well-being. We hypothesized that the association between the variables would be negative. The hypothesis of a significant association between the variables was confirmed. We found that well-being is negatively weakly associated with vaccination beliefs and the four factors: perceived susceptibility and perceived severity, perceived benefits, perceived barriers, and cues to action. That is, as an individual’s concerns about risk and health decrease, well-being decreases. These results are indirectly consistent with the findings of Skalski and colleagues [39,40,41] that anxiety regarding COVID-19 is negatively related to psychological well-being. On the other hand, according to the HBM, it is the threats and fears assessed by this scale that may increase mobilization for vaccination.

Beliefs about vaccination are one of the main barriers to vaccine uptake and the achievement of herd immunity, especially when protecting the most vulnerable populations [6,7]. The HBM is widely used to gain accurate insights into people’s health behaviours, and plays an important role as a model to diagnose beliefs and attitudes about vaccine uptake. It has been used in many previous studies regarding vaccination [18,23,26]. Therefore, examining the relevant dimensions of the HBM in relation to the COVID-19 vaccine seemed important for individualized interventions aimed at increasing acceptance of the vaccine when it becomes available. Among the interventions recommended in the light of the above results and the current pandemic situation are, first of all, strengthening the decision-making competence of undecided persons by offering them the fullest possible information on the advantages and risks of the subject of their decision and defining the risk factors precisely. It is also necessary to make social policy more flexible in order to respond to new facts, such as momentary doubts about the use of a particular vaccine, and to be cautious about giving any preference to vaccinated persons. Immunization is an important health-promoting behaviour to improve public health, and the development of an effective and safe COVID-19 vaccine is expected to reduce morbidity and mortality due to COVID-19. Therefore, it is important to diagnose beliefs about COVID-19 vaccination.

It should be noted that the data obtained on attitudes toward COVID-19 vaccination relate only to the assumptions of the HBM. It seems interesting from the point of view of securing public health to look for other (not included in the HBM) predictors of pro-vaccine attitudes. Since Poland is one of the most Catholic countries in the world (affiliation to the Roman Church is declared by about 80 percent of the population), it seems interesting in future research to determine the possible influence of the authority of the priest (in addition to the observed authority of the doctor) on decision-making in vaccinating themselves and their children. 

Finally, studies have demonstrated that coercive measures, such as the mandatory use of a vaccine passport, can be interpreted as a threat to human rights and civil liberty and consequently reduce vaccine acceptance [42]. Therefore, interventions that support individuals’ autonomous motivation to vaccinate should be implemented alongside developed laws, even during the non-pandemic period. The development of national public health laws and policies to ensure a proportionate and gradual approach to mandatory vaccination in the context of a global health crisis, in addition to information campaigns, can be an effective strategy for infectious disease preparedness and prevention [42,43].

## 5. Limitations

The study is correlational in nature and the data are collected from self-reports, so caution should be used when interpreting the results. Future work on the scale will also require in-depth analyses of the questionnaire validity. The high proportion of young participants represents a limitation on the external validity of our study. This is also indicated by the resulting age effect on the overall vaccination belief score, the subscales of perceived susceptibility, and perceived severity and cues to action. Generalizability of the results is also substantially limited by the fact that this is a convenience sample. We did not measure absolute stability in the validation study. We assumed that the controlled aspects should be considered in terms of intensity, which changes under external influences. In addition, we would like to note that in the study we did not verify the actual lack of vaccination against COVID-19 among the participants, but only based on the declarations in this regard. Participants were included in the study by email invitation for those who attended the University of Bialystok in northeastern Poland. Therefore, the final sample is not representative of the Polish population.

In addition, the part of the survey was conducted using a non-standard questionnaire. These were questions about the sociodemographic characteristics of the respondents. In future research, it is useful to assess the socioeconomic status (SES) of respondents. There are three key measures most commonly used to capture SES in most studies: income, education, and occupation.

Although these data require further exploration, they allow us to highlight some aspects useful for improving vaccine promotion strategies in this particular population.

## 6. Conclusions

In this work, we conducted a preliminary study to evaluate the structure of the questionnaire. The Health Belief Scales Toward COVID-19 Vaccine demonstrated good psychometric properties. It proved to be a valid and reliable tool to assess attitudes towards vaccination in four dimensions consistent with the HBM. The data obtained have useful value. They can be a helpful tool for health educators to assess attitudes about vaccination when new vaccines need to be implemented before they are more widely used in the community. They can also be used in public policy and in the development of education and prevention programmes.

## Figures and Tables

**Figure 1 ijerph-19-05424-f001:**
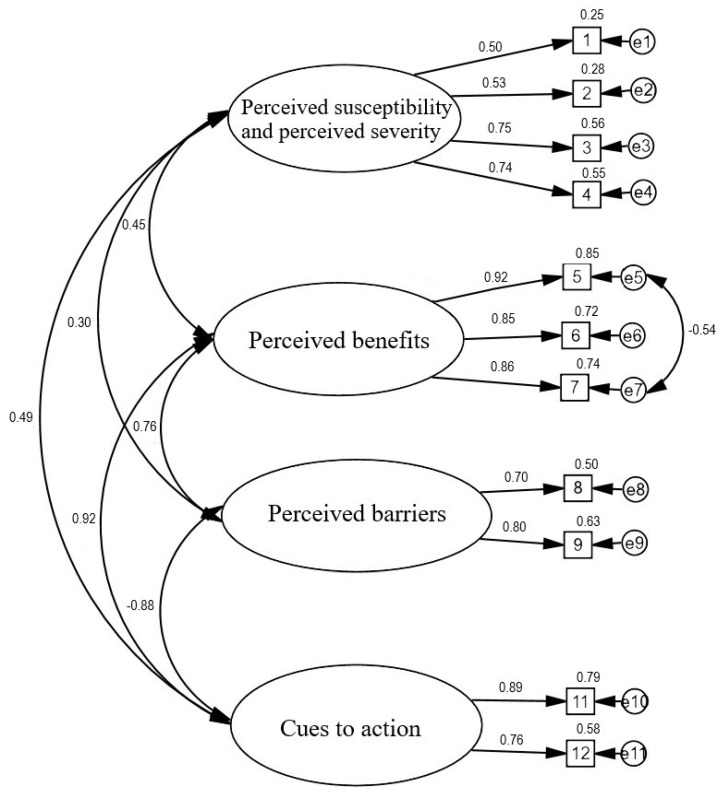
Confirmatory factor analysis for the HBM scale.

**Table 1 ijerph-19-05424-t001:** Further demographic characteristics of the sample.

Variable	*n*	%
**Financial Situation**		
Very good	57	12
Good	186	39
Average	136	29
Bad	79	17
Very bad	14	3
**Place of Residence**		
Urban	328	70
Rural	144	30
**Work Situation**		
Gainfully employed	166	35
Only studying	118	25
Gainfully employed and studying	160	34
Unemployed	28	6

*n* = sample size (subsample); % = percentage based on total answers.

**Table 2 ijerph-19-05424-t002:** Descriptive statistics of the items in Polish and English languages.

	Min	Max	M	SD	S	K
1. Jestem w grupie wysokiego ryzyka zakażenia się koronawirusem. I am at high risk of COVID-19 infection.	1	5	2.06	1.22	0.948	−0.139
2. Myślę, że w najbliższej przyszłości zachoruje na COVID-19. I think I will get COVID-19 in the near future.	1	5	2.45	1.10	0.299	−0.639
3. Uważam, że mógłbym być ciężko chory, gdybym zachorował na COVID-19. I could be severely ill if I got COVID-19.	1	5	2.28	1.12	0.697	−0.143
4. Boje się na samą myśl o możliwości zachorowania na COVID-19. I am afraid of even think about getting illness with COVID-19.	1	5	2.15	1.25	0.835	−0.461
5. Szczepienie uchroni mnie przed zachorowaniem na COVID-19. Immunisation will prevent me from contracting COVID-19.	1	5	2.70	1.37	0.124	−1.270
6. Będąc zaszczepionym, uchronię innych przed zakażeniem koronawirusem. By being immunised and not getting illness, I will protect others from COVID-19.	1	5	2.83	1.41	0.048	−1.314
7. Dzięki zaszczepieniu będę mieć mniejsze obawy o ciężki przebieg COVID-19. By being immunised, I feel less worried about possibility of severe illness from getting COVID-19.	1	5	3.35	1.42	-0.402	−1.137
8. Obawiam się, że szczepionka na COVID-19 może powodować działania niepożądane. I am afraid that COVID-19 vaccine can cause AEFIs.	1	5	2.69	1.38	0.255	−1.223
9. Zakażenie koronawirusem może samo ustąpić, a szczepienie nie jest konieczne. COVID-19 infection can be self-limiting and vaccination is unnecessary.	1	5	2.97	1.40	0.059	−1.225
10. Uważam, że w przyszłości aby się zaszczepić koszt szczepionki na COVID-19 będzie wysoki. I think that the cost of COVID-19 vaccine will be expensive.	1	5	3.06	1.30	0.003	−0.982
11. Uważam, że wszyscy powinni się zaszczepić w celu promowania zdrowia publicznego. I think that all people should be vaccinated to promote public health.	1	5	2.84	1.50	0.122	−1.414
12. Zaszczepię się na COVID-19, jeśli lekarz tak zaleci. I will receive a COVID-19 vaccine if my healthcare workers recommended a vaccination.	1	5	3.09	1.51	−0.096	−1.414

Min = minimum; max = maximum, M = mean; SD = standard deviation; S = skewness; K = kurtosis.

**Table 3 ijerph-19-05424-t003:** Descriptive statistics of the Health Belief Scales Toward COVID-19.

	Min	Max	M	SD	S	K
The Health Belief Scales Toward COVID-19	1.00	5.00	2.75	0.97	−0.07	−1.01
Perceived susceptibility and perceived severity	1.00	5.00	2.23	0.87	0.64	0.01
Perceived benefits	1.00	5.00	2.96	1.27	−0.09	−1.13
Perceived barriers	1.00	5.00	2.83	1.23	0.13	−1.08
Cues to action	1.00	5.00	2.97	1.38	−0.01	−1.29

Min = minimum; max = maximum; M = mean; SD = standard deviation; S = skewness; K = kurtosis.

## Data Availability

The data presented in this study are available on request from the corresponding author.

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
