# Peer review of "Development and Psychometric Properties of the Health Belief Scales toward COVID-19 Vaccine: A Cross-Sectional Study in North-Eastern Poland"

_ijerph, 2022, doi:10.3390/ijerph19095424_

Round 1
Reviewer 1 Report
This is a well-written paper. I think that it is ready to be published after minor changes.
Methods
1. You should explain why did you choose a specific group of participants who did not get a vaccine before November 2021 (almost one year after COVID vaccines introduction)
2. In Table 1, you should provide translated into English statements.
Author Response
This is a well-written paper. I think that it is ready to be published after minor changes.
Thank you very much for your positive feedback.
Methods
1. You should explain why did you choose a specific group of participants who did not get a vaccine before November 2021 (almost one year after COVID vaccines introduction)
The following sentences were added to the "Data Analysis" and "Introduction" sections:"
We hypothesized that the association between the variables would be negative. The study sought to answer the following research question: What are the psychometric properties of the Polish version of the Health Belief Scales Toward COVID-19 Vaccine?. The data obtained on attitudes related to COVID-19 vaccination in the unvaccinated population may contribute to the development of methods of influence in changing these attitudes and securing health”.
“The University of Bialystok database was used to send information about the study. The database contained e-mails to 12000 students and graduates who agreed to have their e-mail addresses processed for research purposes. In order to obtain reliable information on barriers to vaccination, the criterion for entry into the study was the declaration that one is an unvaccinated person. The survey was conducted on non-vaccinators because it seems crucial to know their views in developing educational implications aimed at health prevention.”
- In Table 1, you should provide translated into English statements.
We have added items in English in Table 1.
Reviewer 2 Report
The manuscript “Vaccinations in Poland: Development and Psychometric Properties of the Health Belief Scales Toward COVID-19 Vaccine” applied one Health Belief Model to measure people’s attitude towards COVID-19 vaccine.
There are multiple major defects in this study. First, the study design was not solid, which solely based on online questionnaire—google forms platform. The ethical consent was missing in this study. Even if this study was carefully proceeded, analysis of gender, age, health factors was lacking, considering 472 people participated in this study. Another defect of this manuscript is that this HBM wasn’t clearly described in the method part. Is this model reliable for a pandemic like COVID-19? Why didn’t compare the online data with real-world vaccination data? In addition, there is a language barrier between Polish and English, with multiple Polish phrases in the main text part. How could the author guarantee understand these translated questions well?
Author Response
First, the study design was not solid, which solely based on online questionnaire—google forms platform. The ethical consent was missing in this study.
1. We obtained the consent of the Ethics Committee of the Institute of Psychology of the Polish Academy of Sciences. This was indicated in the first version of the manuscript:
“The study was conducted in November and December 2021 with the approval of the Ethics Committee of the Institute of Psychology, Polish Academy of Science. The participants were informed that: (1) participation in the study was voluntary; (2) they could decline to participate at any time during the study; (3) the study was anonymous; (4) individual responses would not be shared with anyone; and (5) aggregate results would be analysed and used for research purposes only. The Google Forms platform was used to collect data. Respondents gave informed consent and were informed of the objectives and procedure. All procedures performed in the study involving the human participants complied with the ethical standards of the Ethics Committee of the Polish Academy of Science.”
Even if this study was carefully proceeded, analysis of gender, age, health factors was lacking, considering 472 people participated in this study.
2. We have added the following paragraph in section " Descriptive statistics of the Health Belief Scales Toward COVID-19 Vaccine and relations based on demographic variables":
To complete the data analysis, calculations were performed on the descriptive statis-tics for the COVID-19 vaccine belief total score and the four subscales. These results are summarized in Table 3. Using Student’s t-test analysis for independent samples, we ex-amined whether there was a relationship between the COVID-19 vaccination belief score we had created and the sex of the respondent. We confirmed that there were differences between women on the perceived susceptibility and perceived severity index (t(470) = -2.45, p < 0.05; Cohen’s d = 0.24). For the other factors, no statistically significant differences were found between men and women. People from rural areas had higher level of perceived barriers index than people from cities (t(470) = -2.04, p < 0.05; Cohen's d = 0.20). Place of residence did not differentiate the other indicators in a statistically significant manner.
One-way analysis of variance (ANOVA) did not confirm that financial situation and work situation differentiated the results for beliefs about receiving the COVID-19 vaccine and the four component factors of these beliefs in a statistically significant manner.
The overall vaccination belief score was associated with age (r = 0.13; p < 0.01). This correlation was positive and weak. Also, the correlations between age and perceived sus-ceptibility and perceived severity (r = 0.15; p < 0.01) and age and cues to action (r = 0.09; p < 0.05) were weak and positive. However, there were no significant associations between age and perceived benefits or age and perceived barriers.
Another defect of this manuscript is that this HBM wasn’t clearly described in the method part. Is this model reliable for a pandemic like COVID-19? Why didn’t compare the online data with real-world vaccination data? In addition, there is a language barrier between Polish and English, with multiple Polish phrases in the main text part. How could the author guarantee understand these translated questions well?
3. The following sentence has been added to the sections introduction data analysis:
"People of all ages can become infected with the virus that causes COVID-19 disease, and the elderly and people with chronic illnesses are at high risk for severe illness and death. Acceptance and adoption of new vaccines is a significant and unprecedented challenge. In previous studies, the HBM has proven to be a useful tool for predicting vaccination behavior [10-22]."
”We hypothesized that the association between the variables would be negative. The study sought to answer the following research question: What are the psychometric properties of the Polish version of the Health Belief Scales Toward COVID-19 Vaccine?. The data obtained on attitudes related to COVID-19 vaccination in the unvaccinated population may contribute to the development of methods of influence in changing these attitudes and securing health”.
“The University of Bialystok database was used to send information about the study. The database contained e-mails to 12000 students and graduates who agreed to have their e-mail addresses processed for research purposes. In order to obtain reliable information on barriers to vaccination, the criterion for entry into the study was the declaration that one is an unvaccinated person. The survey was conducted on non-vaccinators because it seems crucial to know their views in developing educational implications aimed at health prevention.”
4. We have added items in English in Table 1.
Reviewer 3 Report
Below is my review of the manuscript: "Vaccinations in Poland: Development and Psychometric Properties of the Health Belief Scales Toward COVID-19 Vaccine"
Introduction
Unfortunately, the situation related to Covid-19 is changing very dynamically. Therefore, the authors should update some of the information from the introduction or write exactly that the data provided refer to the period in which the study was conducted. E.g. lines 44-45. According the information Raport szczepień przeciwko COVID-19 - Szczepienie przeciwko COVID-19 - Portal Gov.pl (www.gov.pl) 22 102 872 are fully vaccinated (20.02.2022), that is around 58% of Polish population.
Lines 46-48 need reference.
Lines 76-79 need reference.
Please provide your research questions at the end of the introduction section.
Materials and Methods
“Psychometric properties were assessed on a general sample of 472 Polish participants 109 aged between 16 and 69 years” – People 16-17 have to have their parents' consent. Did you obtain it?
Lines 122-123 “ The invitation to participate in the study was sent to full time and part-time students and graduates of the University of BiaÅ‚ystok” – if so, how it is possible you have in the sample very young people (16-17) and old people (60+)? To how many people the invitation was sent? What was the response rate? Please define what kind of sample it was? Has the lack of vaccination been somehow verified? If not please add it to the limitation section.
Please provide the table of the socio-demographic breakdown of your sample.
Are the questionnaire in Polish and the database publicly available?
Results
Please mind that Table 1 is in the Polish language, you have to provide the translation in the brackets. You do not need min and max here. The information under the table what was the min and max will be enough.
Figure 2 – this figure needs a good legend.
Discussion
I think the discussion is a bit shallow.
How is Polish society different from the other societies in dealing with the pandemic? I would show the problems of researching different societies with one tool. Perhaps there are specific aspects of trust in medicine or trust in the pharmaceutical industry in Poland? It is the case to show the drawbacks of what you have done.
Limitation
Lines 308-309 “The generalizability of the results is therefore limited to the young adult population, and further validation in other adult samples is required”. There is great inconsistency in the whole manuscript about the people you surveyed! You cannot say “16 and 69 years” is the young adult population! Please give a good definition of the sample you have and then you can discuss why you can not generalize.
Author Response
Introduction:
We added literature, research questions, and information.
We added the following paragraphs:
We hypothesized that the association between the variables would be negative. The study sought to answer the following research question: What are the psychometric properties of the Polish version of the Health Belief Scales Toward COVID-19 Vaccine?. The data obtained on attitudes related to COVID-19 vaccination in the unvaccinated population may contribute to the development of methods of influence in changing these attitudes and securing health”.
"People of all ages can become infected with the virus that causes COVID-19 disease, and the elderly and people with chronic illnesses are at high risk for severe illness and death. Acceptance and adoption of new vaccines is a significant and unprecedented challenge. In previous studies, the HBM has proven to be a useful tool for predicting vaccination behavior [10-22]."
We also added the following footnote:
"Data shown are for the period in which the study was conducted, source: https://www.who.int/covid-19/vaccines [accessed 01/20/2022]."
Materials and Methods:
The following paragraphs have been added:
”We hypothesized that the association between the variables would be negative. The study sought to answer the following research question: What are the psychometric properties of the Polish version of the Health Belief Scales Toward COVID-19 Vaccine?. The data obtained on attitudes related to COVID-19 vaccination in the unvaccinated population may contribute to the development of methods of influence in changing these attitudes and securing health”.
“The University of Bialystok database was used to send information about the study. The database contained e-mails to 12000 students and graduates who agreed to have their e-mail addresses processed for research purposes. In order to obtain reliable information on barriers to vaccination, the criterion for entry into the study was the declaration that one is an unvaccinated person. The survey was conducted on non-vaccinators because it seems crucial to know their views in developing educational implications aimed at health prevention.”
We have added a table with group charactistics and the following paragraph:
“Participants were differentiated according to the declaration of financial situation (12% very good, 40% good, 29% average, 16% bad and 3% very bad) and place of residence (30% rural, 70% urban). Among the survey participants 36% were gainfully employed, 34% were gainfully employed and studying, 25% were only studying and 6% were unemployed. All data obtained from respondents were complete. Demographic characteristics of the sample are detailed in Table 1.”
The questionnaire and database are available from the authors.
We added a sentence in the limitations:
"In addition, we would like to note that in the study we did not verify the actual lack of vaccination against COVID-19 among the participants, but only based on the declarations in this regard."
Results
We have added item translations and information under the tables.
Discussion
We have added the following paragraph in the "discussion" section:
"Finally, it should be noted that the data obtained on attitudes toward COVID-19 vaccination relate only to the assumptions of the HBM. It seems interesting from the point of view of securing public health to look for other (not included in the HBM) predictors of pro-vaccine attitudes. Since Poland is one of the most Catholic countries in the world (affiliation to the Roman Church is declared by about 80 percent of the population), it seems interesting in future research to determine the possible influence of the authority of the priest (in addition to the observed authority of the doctor) on decision-making in vaccinating themselves and their children."
Limitations:
We have rewritten the " limitations" sections:
”Generalizability of the results is also substantially limited by the fact that this is a convenience sample. We did not measure absolute stability in the validation study. We assumed that the controlled aspects should be considered in terms of intensity, which changes under external influences. In addition, we would like to note that in the study we did not verify the actual lack of vaccination against COVID-19 among the participants, but only based on the declarations in this regard.”
Reviewer 4 Report
Konaszewski and colleagues have put together a well written paper assessing the psychometric properties of adapted health belief models on COVID-19 vaccine in the Polish population. The authors well described the procedures of polish adaptation of the scales as well as detailed mythologies assessing structure, reliability, and validity of this scale. The general analysis is valid but have several comments below:
Major concerns:
1) As the authors point out, the objective of this study is a Polish adaptation of HBM-based tool, so I am wondering if author could provide their systemic sampling strategy representing the characteristics of general Polish population. I believe participant’s characteristics table should be reported like age, gender, education, or SES distribution in this sample. It should be cautious to address “Polish adaptation” tool without a big sample size or systematic random sampling strategy.
2) The author described age of participants was between 16-69 but the mean value was around 25, suggesting age distribution might be skewed, and it should be cautious to run parametric test (person or t test) if normality not reach or extreme outliers exist on your variables. In Line 238-241, person correlation is good if meet assumption but I would suggest using linear regression to see the effect of age (or other demographic factors like education) on vaccine belief scores.
Minor:
1) Should add explanations under each table. (like M=mean or median, SD= Standard deviations….)
Author Response
Thank you very much for your positive feedback.
Major concerns:
1) As the authors point out, the objective of this study is a Polish adaptation of HBM-based tool, so I am wondering if author could provide their systemic sampling strategy representing the characteristics of general Polish population. I believe participant’s characteristics table should be reported like age, gender, education, or SES distribution in this sample. It should be cautious to address “Polish adaptation” tool without a big sample size or systematic random sampling strategy.
1) The following sentence has been added to the sections introduction data analysis:
”We hypothesized that the association between the variables would be negative. The study sought to answer the following research question: What are the psychometric properties of the Polish version of the Health Belief Scales Toward COVID-19 Vaccine?. The data obtained on attitudes related to COVID-19 vaccination in the unvaccinated population may contribute to the development of methods of influence in changing these attitudes and securing health”.
“The University of Bialystok database was used to send information about the study. The database contained e-mails to 12000 students and graduates who agreed to have their e-mail addresses processed for research purposes. In order to obtain reliable information on barriers to vaccination, the criterion for entry into the study was the declaration that one is an unvaccinated person. The survey was conducted on non-vaccinators because it seems crucial to know their views in developing educational implications aimed at health prevention.”
1) We have added a table with group charactistics and the following paragraph: “Participants were differentiated according to the declaration of financial situation (12% very good, 40% good, 29% average, 16% bad and 3% very bad) and place of residence (30% rural, 70% urban). Among the survey participants 36% were gainfully employed, 34% were gainfully employed and studying, 25% were only studying and 6% were unemployed. All data obtained from respondents were complete. Demographic characteristics of the sample are detailed in Table 1.”
2) The author described age of participants was between 16-69 but the mean value was around 25, suggesting age distribution might be skewed, and it should be cautious to run parametric test (person or t test) if normality not reach or extreme outliers exist on your variables. In Line 238-241, person correlation is good if meet assumption but I would suggest using linear regression to see the effect of age (or other demographic factors like education) on vaccine belief scores.
2) We have added the following paragraph in section " Descriptive statistics of the Health Belief Scales Toward COVID-19 Vaccine and relations based on demographic variables":
"To complete the data analysis, calculations were performed on the descriptive statistics for the COVID-19 vaccine belief total score and the four subscales. These results are summarized in Table 3. Using Student’s t-test analysis for independent samples, we ex-amined whether there was a relationship between the COVID-19 vaccination belief score we had created and the sex of the respondent. We confirmed that there were differences between women on the perceived susceptibility and perceived severity index (t(470) = -2.45, p < 0.05; Cohen’s d = 0.24). For the other factors, no statistically significant differences were found between men and women. People from rural areas had higher level of perceived barriers index than people from cities (t(470) = -2.04, p < 0.05; Cohen's d = 0.20). Place of residence did not differentiate the other indicators in a statistically significant manner.
One-way analysis of variance (ANOVA) did not confirm that financial situation and work situation differentiated the results for beliefs about receiving the COVID-19 vaccine and the four component factors of these beliefs in a statistically significant manner.
The overall vaccination belief score was associated with age (r = 0.13; p < 0.01). This correlation was positive and weak. Also, the correlations between age and perceived sus-ceptibility and perceived severity (r = 0.15; p < 0.01) and age and cues to action (r = 0.09; p < 0.05) were weak and positive. However, there were no significant associations between age and perceived benefits or age and perceived barriers."
Minor:
1) Should add explanations under each table. (like M=mean or median, SD= Standard deviations….)
1) We have added explanations under each table.
Reviewer 5 Report
This study “Vaccinations in Poland: Development and Psychometric Properties of the Health Belief Scales Toward COVID-19 Vaccine”, conducted by Konaszewski et al, evaluated the psychometric properties of the Health Belief Scales Toward COVID-19 Vaccine in Poland. A total of 472 polish participants were assessed.
Comment
- The article title “Vaccinations in Poland: Development and Psychometric Properties of the Health Belief Scales Toward COVID-19 Vaccine” is not clear.
- Page 3, 2. Materials and Methods, can the authors explain how they decided the sample size? In addition, up to 69% of the participants were women who had not received a COVID-19 vaccination. Authors should comment on whether enrolling this study group introduces bias.
- It has been known that the characterizations and social background of the enrolled participants can affect the study result. Authors should include those data and analyze further.
- Table 1 should include English translation.
Author Response
This study “Vaccinations in Poland: Development and Psychometric Properties of the Health Belief Scales Toward COVID-19 Vaccine”, conducted by Konaszewski et al, evaluated the psychometric properties of the Health Belief Scales Toward COVID-19 Vaccine in Poland. A total of 472 polish participants were assessed.
Comment
1. The article title “Vaccinations in Poland: Development and Psychometric Properties of the Health Belief Scales Toward COVID-19 Vaccine” is not clear.
1. We have changed the title: “Development and Psychometric Properties of the Health Belief Scales Toward COVID-19 Vaccine in Poland”
2. Page 3, 2. Materials and Methods, can the authors explain how they decided the sample size? In addition, up to 69% of the participants were women who had not received a COVID-19 vaccination. Authors should comment on whether enrolling this study group introduces bias.
2.We added the following paragraphs in the "materials and method" section:
“The University of Bialystok database was used to send information about the study. The database contained e-mails to 12000 students and graduates who agreed to have their e-mail addresses processed for research purposes. In order to obtain reliable information on barriers to vaccination, the criterion for entry into the study was the declaration that one is an unvaccinated person. The survey was conducted on non-vaccinators because it seems crucial to know their views in developing educational implications aimed at health prevention.”
“A "sampling calcuator" was used to calculate the sample size. A 95% confidence level, a fraction size of 0.5, and a maximum error of 5% were used. The total number of respondents is 12000. Analysis with the abovementioned parameters would demand a sample of at least 372 participants.”
3. It has been known that the characterizations and social background of the enrolled participants can affect the study result. Authors should include those data and analyze further.
3. We have added the following paragraph in section " Descriptive statistics of the Health Belief Scales Toward COVID-19 Vaccine and relations based on demographic variables":
To complete the data analysis, calculations were performed on the descriptive statis-tics for the COVID-19 vaccine belief total score and the four subscales. These results are summarized in Table 3. Using Student’s t-test analysis for independent samples, we ex-amined whether there was a relationship between the COVID-19 vaccination belief score we had created and the sex of the respondent. We confirmed that there were differences between women on the perceived susceptibility and perceived severity index (t(470) = -2.45, p < 0.05; Cohen’s d = 0.24). For the other factors, no statistically significant differences were found between men and women. People from rural areas had higher level of perceived barriers index than people from cities (t(470) = -2.04, p < 0.05; Cohen's d = 0.20). Place of residence did not differentiate the other indicators in a statistically significant manner.
One-way analysis of variance (ANOVA) did not confirm that financial situation and work situation differentiated the results for beliefs about receiving the COVID-19 vaccine and the four component factors of these beliefs in a statistically significant manner.
The overall vaccination belief score was associated with age (r = 0.13; p < 0.01). This correlation was positive and weak. Also, the correlations between age and perceived sus-ceptibility and perceived severity (r = 0.15; p < 0.01) and age and cues to action (r = 0.09; p < 0.05) were weak and positive. However, there were no significant associations between age and perceived benefits or age and perceived barriers.
We also added the following paragraph and table with group characteristics:
“Participants were differentiated according to the declaration of financial situation (12% very good, 40% good, 29% average, 16% bad and 3% very bad) and place of residence (30% rural, 70% urban). Among the survey participants 36% were gainfully employed, 34% were gainfully employed and studying, 25% were only studying and 6% were unemployed. All data obtained from respondents were complete. Demographic characteristics of the sample are detailed in Table 1.”
4.Table 1 should include English translation.
4. We have added items in English in Table 1.
Reviewer 6 Report
This is an interesting and useful manuscript which reports the testing/validation of the "Health Belief Scales toward COVID19 Vaccine" in Poland.
Overall, I think the manuscript is generally thorough and reasonably well presented. I do have the following issues:
- A little more information on who was sent the invitation for the survey is needed (lines 121-122): were all current undergraduate students and all graduates of the university contacted? Or was a sample contacted? If the latter, how was the sample selected?
- Table 1: It would be very helpful to have each of these questions given in English as well.
- Also Table 1: Are the "S" and "K" meant to represent skew? Kurtosis?The statistic labels should be indicated somewhere in the table.
- Since the WHO 5 item well being index is only 5 items, please list all five of these in the description of the measure. (lines 151-156)
- Lines 251-256 on the scoring of the Health Belief measure should be put where the measure is described (lines 141-150). Additional information should also be included when describing the measure: how was a total score constructed (were items added once appropriate reverse coding was used)? How were missing data addressed?
- Figure 1: Should read "confirmatory" rather than just "confirm"
- Lines 274-283: The expectations about relationship between the Health Beliefs and the WHO measure should be discussed earlier. Hypotheses should be stated prior to analysis. Also, how is this a test of validity (lines 244-250)? How should they be correlated to show this? What kind of validity is meant here? Predictive? Construct?
- Limitations: Lines 308-310--Generalizability of the results is also substantially limited by the fact that this is a convenience sample.
- Limitations: Lines 310-313: This is unclear.
Author Response
1. A little more information on who was sent the invitation for the survey is needed (lines 121-122): were all current undergraduate students and all graduates of the university contacted? Or was a sample contacted? If the latter, how was the sample selected?
1. We added the following paragraphs in the "materials and method" section:
“The University of Bialystok database was used to send information about the study. The database contained e-mails to 12000 students and graduates who agreed to have their e-mail addresses processed for research purposes. In order to obtain reliable information on barriers to vaccination, the criterion for entry into the study was the declaration that one is an unvaccinated person. The survey was conducted on non-vaccinators because it seems crucial to know their views in developing educational implications aimed at health prevention.”
2. Table 1: It would be very helpful to have each of these questions given in English as well.
2. We have added items in English in Table.
3. Also Table 1: Are the "S" and "K" meant to represent skew? Kurtosis?The statistic labels should be indicated somewhere in the table.
3. We have added explanations under each table.
4.Since the WHO 5 item well being index is only 5 items, please list all five of these in the description of the measure. (lines 151-156)
4. We have added items: 'I have felt cheerful and in good spirits'; 'I have felt calm and relaxed'; 'I have felt active and vigorous'; 'I woke up feeling fresh and rested'; and 'My daily life has been filled with things that interest me’
5. Lines 251-256 on the scoring of the Health Belief measure should be put where the measure is described (lines 141-150). Additional information should also be included when describing the measure: how was a total score constructed (were items added once appropriate reverse coding was used)? How were missing data addressed?
5. We added the following sentences:
“All data obtained from respondents were complete.” and “In addition, the study can determine an overall scale index by summing the scores obtained across all statements. … The scale is based on the HBM model, which has been shown to be effective in predicting COVID-19 vaccination attitudes in the Vietnamese population (see [26]).”
6. Figure 1: Should read "confirmatory" rather than just "confirm"
6. We have renamed Figure 1.
7. Lines 274-283: The expectations about relationship between the Health Beliefs and the WHO measure should be discussed earlier. Hypotheses should be stated prior to analysis. Also, how is this a test of validity (lines 244-250)? How should they be correlated to show this? What kind of validity is meant here? Predictive? Construct?
7. We added the following sentences and information: Convergent Validity
"Validity was assessed by evaluating the correlation coefficients with the WHO-5 score. We expected, a negative correlation between the constructs."
8. Limitations: Lines 308-310--Generalizability of the results is also substantially limited by the fact that this is a convenience sample. Limitations: Lines 310-313: This is unclear.
8. We have rewritten the " limitations" sections:
”Generalizability of the results is also substantially limited by the fact that this is a convenience sample. We did not measure absolute stability in the validation study. We assumed that the controlled aspects should be considered in terms of intensity, which changes under external influences. In addition, we would like to note that in the study we did not verify the actual lack of vaccination against COVID-19 among the participants, but only based on the declarations in this regard.”
Round 2
Reviewer 2 Report
HBM is critical for this study. Suggest the author provide enough description about this tool.
Author Response
HBM is critical for this study. Suggest the author provide enough description about this tool.
Thank you for your suggestions. We have added more information about the scale.
Reviewer 3 Report
No comments.
Author Response
Thank you very much for your positive feedback.